# Temperature Dependence of the Beating Frequency of hiPSC-CMs Using a MEMS Force Sensor

**DOI:** 10.3390/s23073370

**Published:** 2023-03-23

**Authors:** Ryota Ikegami, Takuya Tsukagoshi, Kenei Matsudaira, Kayoko Hirayama Shoji, Hidetoshi Takahashi, Thanh-Vinh Nguyen, Takumi Tamamoto, Kentaro Noda, Ken’ichi Koyanagi, Toru Oshima, Isao Shimoyama

**Affiliations:** 1Department of Intelligent Robotics, Faculty of Engineering, Toyama Prefectural University, Imizu 939-0398, Japan; 2Department of Mechano-Informatics, Graduate School of Information Science and Technology, The University of Tokyo, Tokyo 113-8654, Japan; 3Department of Mechanical Engineering, Faculty of Science and Technology, Keio University, Yokohama 223-8522, Japan; 4Sensing System Research Center, National Institute of Advanced Industrial Science and Technology (AIST), Tsukuba 305-8564, Japan; 5Department of Intelligent Mechanical Engineering, Faculty of Engineering, Fukuoka Institute of Technology, Fukuoka 811-0295, Japan; 6Toyama Prefectural University, Imizu 939-0398, Japan; 7The University of Tokyo, Tokyo 113-8654, Japan

**Keywords:** piezoresistive cantilever, cardiomyocytes, temperature dependence

## Abstract

It is expected that human iPS cell-derived cardiomyocytes (hiPSC-CMs) can be used to treat serious heart diseases. However, the properties and functions of human adult cardiomyocytes and hiPSC-CMs, including cell maturation, differ. In this study, we focused on the temperature dependence of hiPSC-CMs by integrating the temperature regulation system into our sensor platform, which can directly and quantitatively measure their mechanical motion. We measured the beating frequency of hiPSC-CMs at different environmental temperatures and found that the beating frequency increased as the temperature increased. Although the rate at which the beating frequency increased with temperature varied, the temperature at which the beating stopped was relatively stable at approximately 20 °C. The stopping of beating at this temperature was stable, even in immature hiPSC-CMs, and was considered to be a primitive property of cardiomyocytes.

## 1. Introduction

Cardiomyocytes derived from human stem cells have high potential in applications such as regenerative medicine and disease modeling [1,2,3,4,5]. The specific immaturity of hiPSC-CMs includes irregularities in the sarcomere structure, the contractile mechanism of cardiomyocytes, and the presence of autopotency [6,7]. Although it has been reported that the maturation of hiPSC-CMs was enhanced by using electrical stimulation or by culturing the cells in a 3D environment, the immaturity of hiPSC-CMs still remains a major challenge in the medical field [8,9,10,11,12].

In order to bring hiPSC-CM-related technology into clinical practice, it is necessary to quantitatively evaluate the functional differences between hiPSC-CMs and mature cardiomyocytes. As the function of the heart is to pump blood in the body, it is critical to quantitatively evaluate the beating properties of immature hiPSC-CMs and adult cardiomyocytes. However, most of the previously reported methods, such as micropost arrays [13,14,15] and fluorescent beads [13,16,17] based on microscopic image processing, are not suitable for studying cardiomyocyte motion in real time with high temporal resolution. Therefore, we previously built a system that uses a piezoresistive force sensor to measure the contractile force of hiPSC-CMs [18]. To quantitatively evaluate the contractile force of cardiomyocytes, its force must be measured with high temporal resolution. It is also preferable to measure it in a mechanical environment similar to that in vivo. In detail, we applied a stretch stimulus to hiPSC-CMs and measured the force–length relationship with a MEMS force sensor to evaluate the relationship between the stimulus and the contractile force [19]. In turn, we used the measured force to induce isometric and auxotonic contractions in hiPSC-CMs by controlling the strength of the stretch stimulus [20].

Silicon-based MEMS force sensors have been used in many applications due to their high sensitivity and ease of mass production. Recently, designs have been proposed that not only improve their sensitivity and responsiveness, but also provide robustness against temperature changes at the sensor level [21]. In this previous study, stable microflow measurement was achieved by improving the performance of temperature compensation. It was shown that force sensors based on the piezoresistive effect could be fabricated using graphene as well as silicon [22]. Force sensors based on graphene also have high sensitivity (nN-order) that is not inferior to that of silicon. Piezoresistive devices based on amorphous carbon have also been proposed and are expected to be flexible and reproducible strain sensors [23]. Guo et al. showed that the gauge factor can be controlled by applying a substrate bias voltage to a force sensor made of amorphous carbon [24]. Multiaxial force sensors that measure force as a vector are also being developed. A compact cross-shaped triaxial piezoresistive force sensor has been proposed, which was also shown to be capable of high linearity and low crosstalk [25].

In this study, we investigated how the beating of hiPSC-CMs changes with temperature by measuring their beating with a MEMS force sensor. It is well known that the heart changes its behavior in response to temperature [26]. We implemented a temperature-regulating system in the sensor system described above and measured the pulsation of hiPSC-CMs at different environmental temperatures. It was found that the pulsation frequency decreased as the temperature decreased. Although the temperature coefficient of beating frequency varied widely among 11 experiments (0.026–0.16 Hz/deg), the beating frequency approached zero and stopped at around 20 °C in all cell groups. This pulsation arrest temperature is common to the behavior of the heart in non-hibernating mammals, including humans [27]. Although the arrest of the animal heartbeat in this temperature range has been attributed to the decreased spontaneous depolarization of the sinus node [28,29], the fact that cardiomyocytes themselves have a similar function suggests that this phenomenon originates from a more fundamental origin of the heart.

## 2. Materials and Methods

An overview of the measurement system [19,20] used in this experiment is shown in Figure 1. The system consisted of a piezoresistive cantilever-type force sensor, a movable plate for culturing cells, a piezoelectric stage (B17-090, Nano Control Co., Ltd., Tokyo, Japan) for adjusting the sensor position, a piezo controller (NCS 6121C, Nano Control Co., Ltd., Tokyo, Japan), a field-programmable gate array (FPGA) (cRIO-9030, National Instruments, Austin, TX, USA) for controlling the piezoelectric stage and acquiring data, and a host PC for managing the FPGA and recording data. The system also included a microscope (IX-71, Olympus Corp., Tokyo, Japan), and a heater (INU-UK-B18M, Tokai Hit Corp., Shizuoka, Japan) was installed on the sample stage of the microscope. Therefore, during the experiments, the temperature of the cells and culture medium could be controlled while they were observed under the microscope. The movable plate was fixed to the bottom of a petri dish and immersed in the culture medium with the cells after the cells were seeded. The cantilever-type force sensor was fixed on the piezoelectric stage, and the position of the sensor was adjusted by the piezoelectric stage. The piezoelectric stage had a positioning accuracy of ±2 nm, a moving stroke of 100 µm, and a load capacity of 5 N. The FPGA was controlled by a LabVIEW program (National Instruments, Austin, TX, USA), and the force sensor was controlled via the piezoelectric stage. The position could be adjusted, and the signals from the sensors were recorded.

### 2.1. Movable Plate

The movable plate was fabricated through the MEMS fabrication process and had a movable part and a fixed part. The plate was made of Si and was generally nontoxic to cells. A photograph of the movable plate is shown in Figure 2A, an image of the movable plate is shown in Figure 2B, and the dimensions of the movable plate are shown in Figure 2C. The fixed component was fixed to a petri dish and thus did not move. The movable component was connected to the fixed part through two thin arms and was easily displaced by the pulsation of the cardiomyocytes. A portion of the cardiomyocytes that were seeded on the movable plate grew on the movable and fixed plates. When the cardiomyocytes beat, the movable component was pulled and deformed due to cell movement. The movable part had a protrusion that was in contact with the sensor. When the cell contracted, the movable part was displaced, deforming the sensor and producing a signal. As the contamination of the plate surface inhibited cell adhesion, the fabricated plate surface was cleaned and O_2_ plasma etching was performed for 20 min. As O_2_ plasma etching made the plate’s surface hydrophilic, the plate was left for at least 3 days to allow the surface to become hydrophobic again before it was used in the experiment. The hydrophobic nature of the plate’s surface was useful for effective fibronectin coating prior to cell culturing.

### 2.2. Force Sensor

In this system, a piezoresistive cantilever was produced through the MEMS fabrication process and used as a force sensor in the experiment. The piezoresistive effect was a property in which the electrical resistivity changed with the magnitude of the strain. Therefore, in this experiment, the deformation of the sensor in contact with the protruding part of the plate was acquired as a voltage. A Wheatstone bridge was used to convert the resistance change into a voltage signal. As the voltage value obtained by the sensor was very weak, an amplifier circuit was used to augment the signal. The sensor signal naturally included an offset, and a bridge circuit was connected to the amplifier circuit to remove this offset. The signal amplified by this circuit was acquired by the FPGA connected to the sensor, converted to the displacement or force, and recorded.

The spring constant *k*_s_ of the sensor, the spring constant *k*_p_ of the plate, and the displacement ∆*X* of the sensor can be used to calculate the magnitude of the load *F* on the sensor as [19]:$$F=\left(k_{s}+k_{p}\right)\Delta X.$$

The displacement ∆*X* of the sensor is determined based on the change in the voltage ∆*V*. The voltage changes due to the piezoresistive effect. The relationship between the fractional resistance change ∆*R*/*R* and the change in voltage ∆*V* for a sensor displacement of ∆*X* is:$$\Delta V\;\approx -\frac{1}{4}\frac{\Delta R}{R}.$$

As the relationship between the sensor displacement and the fractional resistance change is obtained from the calibration, the load *F* applied to the sensor can be derived according to the voltage change ∆*V* with the above equation. If the gain of the amplifier is expressed as *G*, the voltage change is given by
$$\Delta V\;\approx -\frac{1}{4}G\frac{\Delta R}{R}.$$

A photograph of the actual sensor is shown in Figure 3. The sensor was fabricated in our lab via the MEMS process. The size of the sensor was 300 µm × 60 µm and its thickness was 2 µm. This sensor was fixed to a piezoelectric stage, and the sensor position could be adjusted by the piezoelectric stage. The piezoelectric stage was connected to a piezoelectric controller and the FPGA. Therefore, the piezoelectric stage could be controlled by the FPGA in real time during measurements. The piezoelectric stage could also maintain a position specified by the PI control of the piezo controller. The piezoelectric stage was used to connect the sensor to the plate. After the sensor came into contact with the plate, the displacement of the piezoelectric stage caused an offset in the sensor signal. This offset was removed by programming the FPGA.

Prior to the measurement of the contractile force of the cardiomyocytes, the sensitivity of the sensor (ratio of the fractional resistance change ∆*R/R* to the amount of displacement ∆*X*) was measured using the following procedure. The piezo stage was controlled by FPGA and the sensor was pressed against a rigid plate with various forces. The relationship between the amount of displacement (estimated from location of the piezostage) and the fractional resistance change of the sensor showed that the sensitivity of the sensor system was 1.18 × 10^−3^ μm^−1^. This means that the voltage value of the sensor signal could be converted into contractile force by using the value of the spring constant, which will be discussed below.

### 2.3. Cultivation of the hiPSC-CMs

hiPSC-CMs (MiraCell, TAKARABIO, Inc., Shiga, Japan) were cultured on coated plates (fibronectin) for more than 5 days before their use in experiments. The plates were sprayed with 70% ethanol the day before seeding and sterilized with UV light for 24 h. On the day of seeding, the plates were again sprayed with 70% ethanol for sterilization and dried under UV light until the ethanol was completely volatilized. Then, 100 µL of 50 µg/mL fibronectin was applied to the plates, and the plates were left in an incubator for 2 h at 37 °C in a 5% CO_2_ atmosphere. This process was performed to increase the activity of the fibronectin and to ensure that the fibronectin adhered to the plate. Then, cardiomyocytes adjusted to approximately 7.0 × 10^6^ cells were seeded with MiraCell CM Thawing Medium (TAKARABIO, Inc., Shiga, Japan). After the cells were seeded, they were incubated in an incubator at 37 °C in a 5% CO_2_ atmosphere for 5 h to promote adhesion to the plate. To prevent contamination, MiraCell CM Culture Medium (TAKARABIO, Inc., Shiga, Japan) with 1% penicillin–streptomycin was added to the culture, which was then incubated at 37 °C in a 5% CO_2_ atmosphere. Half of the medium was changed every 2 days according to the manufacturer’s protocol, and the entire medium was changed the day before measurements were taken. A microscopic image of hiPSC-CMs 6 days after seeding is shown in Figure 4. MiraCell CM Culture Medium (TAKARABIO, Inc., Shiga, Japan) was used in all experiments.

### 2.4. Measurement of Beating Cardiomyocytes at Different Temperatures

To observe differences in the beating of the hiPSC-CMs at different temperatures, the temperature was varied from 24 °C to 40 °C in steps of 4 °C, and the beating of the cells was measured. The cell temperature was controlled using an incubator mounted on the sample stage of the microscope and a tabletop heater. The temperature of the medium was measured using an infrared thermometer to ensure that no noise was introduced to the sensor. The sensors were sterilized with 70% ethanol and ultraviolet light immediately before the experiment. If part of the sensor came into contact with the water’s surface, the surface tension would distort the water’s surface, making microscopic observations impossible. Therefore, 10 mL of medium at 37 °C was added to the petri dish after it was removed from the incubator to ensure that the sensor was completely submerged. After the medium was added, the cells were allowed to cool naturally for 20 min; then, the cells were placed on a tabletop heater, and the sensor was brought into contact with the plate under microscopic observation. The cells adhered to the plate and then beat, so the plate was displaced as the cells beat. To accurately measure the contractile force of the cells, the position of the plate, which was not affected by the beating of the cells, was used as a reference. The data acquisition sampling rate of the FPGA was 1 kHz. The temperature of the culture medium in which the cells were seeded was adjusted with a tabletop heater, which took 10 to 20 min to change the temperature. Then, after the temperature had stabilized sufficiently, the measurement of beating was started. During the temperature adjustment, the sensor was moved away from the plate, and the sensor’s power was turned off to reduce the influence of the sensor on the cells. At each temperature, a one-minute or three-minute beat was recorded. After the experiment, the entire medium was replaced to prevent contamination. The sensors were then sterilized with 70% ethanol and stored on a clean bench.

## 3. Results

Validated measurement results were obtained for 11 cell groups, which were named groups A to K. Figure 5 shows an example of the results obtained by increasing the temperature from 20 °C to 38 °C and measuring the pulsation of group D. The pulsation at a temperature of 24 °C is shown in Figure 5A. A fast Fourier transform (FFT) was applied to the time variation in the plate’s displacement using the numerical analysis software MATLAB (The MathWorks, Inc., Natick, MA, USA), as shown in Figure 5B. Among the spectral components, the range from 0.3 to 10 Hz, shown in red, was considered to be the main component of the pulsation. In particular, there were clear peaks at 0.49 Hz and its integer multiples, which was considered to be the beating frequency and its harmonics. However, as the time variation in the pulsation was not a sinusoidal wave, but a repetition of shorter pulses, the FFT spectrum included frequency components up to approximately 10 Hz; the band below 0.3 Hz was thought to be the abovementioned slow displacement fluctuation, while the band above 10 Hz was thought to be primarily noise. Therefore, the signal from 0.3 to 10 Hz was extracted and analyzed with an inverse FFT using MATLAB (Figure 5C). The pulsatile displacements of the hiPSC-CMs were extracted, and the positions of the peaks could be calculated accurately. The values of the peaks were automatically extracted by the program and were indicated by red circles.

The analysis method discussed here is equivalent to using a bandpass filter in postprocessing and is an effective method for accurately determining the timing of the beat. However, to investigate the magnitude of the displacement, it is necessary to return to the original data in Figure 5A, and the displacement was estimated to be approximately 10 µm at 24 °C. The spring constants of the movable plate and force sensor were 2.54 N/m and 0.72 N/m, respectively, with a combined spring constant of 3.26 N/m [20]. Thus, a displacement of 10 µm corresponded to a force of approximately 33 µN.

Figure 6A–C shows the beating of group D with variations in temperature from 28 °C, 32 °C, and 36 °C. Figure 5C shows a graph in which the main components of the beating motion are extracted from the FFT spectrum and an inverse FFT is performed to ensure that the peaks are accurate. The figure shows that the higher the temperature, the higher the beating frequency. The number of beats during a 60 s measurement period was 29 at 24 °C, 37 at 28 °C, 62 at 32 °C, and 78 at 36 °C. When these values were converted to the beating frequency, the frequency was 0.49 Hz at 24 °C, 0.62 Hz at 28 °C, 1.04 Hz at 32 °C, and 1.31 Hz at 36 °C, indicating that the frequency increased with temperature.

The relationship between the temperature and the beating frequency is plotted for groups A–K in Figure 7A–K. The black circles indicate the measured pulsation frequencies, which had a linear relationship with the temperature. Therefore, the straight line $f\left(T\right)=\alpha \left(T-T_{0}\right)$, which was obtained by fitting the measured values, is shown as a red line on the graph. Although the relationship was linear for individual hiPSC-CMs, the coefficient values varied widely between hiPSC-CMs. The coefficient α and the temperature at which the beating frequency was zero, which we named the pulsation stop temperature, *T*_0_, are listed in Table 1. Although the approximate curves (slopes *α*s) varied widely, the pulsation stop temperature *T*_0_ was 20.7 ± 2.7 °C, which was relatively stable. The degree to which the coefficients of approximate curves varied could be evaluated by the ratio of the standard deviation to the mean. Table 1 shows that the slope α varied widely (0.61), while the pulsation stop temperature $T_{0}$ remained relatively stable (0.13). It is known that the hearts of non-hibernating mammals, including humans, stop beating at approximately 20 °C [27], and the results in Figure 7 and Table 1 are consistent with this finding. Compared with human adult cardiomyocytes, the hiPSC-CMs used in this experiment were significantly immature; thus, the above results imply that stopping beating at temperature *T*_0_ is a primitive feature of cardiomyocytes. The results of this experiment indicate that beating does not stop in the cardiomyocytes of hibernating mammals, such as bears and squirrels; however, further studies are necessary to confirm this prediction.

Finally, the shape of the beating peaks was examined in detail. The beats were not perfectly regular, and their periods varied slightly from beat to beat, even for consecutive beats at the same temperature. Therefore, we defined the period per beat in the manner shown in Figure 8A. An example is shown in Figure 5C, from which a portion of the data was extracted. First, the time of the peak was automatically detected for all beats, and the intermediate time between the peaks was used as the boundary of the period. The period of a peak was defined as the period from the border of the previous peak to the border of the subsequent peak. After the period of the peak was defined, the first 10 peaks at each temperature were superimposed (Figure 8B). The time on the horizontal axis was normalized to the period of each beat, and the intensity on the vertical axis was normalized to the average peak height at each temperature. The higher the temperature, the sharper the peak in real time (Figure 6). However, when the data were normalized by the period, there was little variation with temperature, and there was more variation at the same temperature. Therefore, we averaged 10 pulsation waveforms taken at the same temperature and examined the variation with the temperature in detail (Figure 8C). As the sampling points were scattered during the normalization process, the normalization time was divided into 1000 parts, and the values of the beating intensity at each normalization time were recalculated by linear interpolation. The 10 beating waveforms were then averaged. Within the range of 24 to 36 °C, the normalized beating waveforms were generally consistent, with little difference due to temperature. The first half of the beat was particularly similar; however, in the second half of the beat, the intensity decreased more rapidly at higher temperatures and more slowly at lower temperatures. However, the differences were minor, and more detailed data were required to discuss cell biological considerations.

## 4. Discussion

We used MEMS-based cantilever force sensors to investigate how the beating frequency of hiPSC-CMs changes with temperature. The use of silicon-based force sensors for studying cellular forces has advantages in terms of temporal resolution [18,19,20,30,31]. Silicon-based force sensors have temporal resolutions in the order of GHz, and it has been reported that the force can be measured in units of adhesive spots on cells because of their high measurement speed of 400 kHz [31]. The present study was based on these previous studies, and the kHz-order sampling rate enabled the accurate detection of the timing of the beats.

It is known that, as a mammal’s core temperature decreases, its heart rate gradually decreases, leading to cardiac arrest at approximately 20 °C [32]. This cardiac arrest is attributed to the cessation of the automatic function of the sinus node. In contrast, the hearts of hibernating mammals do not stop beating at temperatures as low as 5 °C. Uchida et al. investigated the temperature dependence of synchronous beating characteristics in the cardiomyocytes of neonatal rats [33]. They found that the temperature sensitivity of the beating properties increased as the cardiomyocytes matured in long-term cultures. In addition, the field potential duration of the hiPSC-CMs increased at low temperatures, and the Ca^2+^ concentration and K^+^ current increased [34]. While the experimental results described in this paper show a strong linearity between the temperature and beating frequency for individual hiPSC-CMs, there were large variations in the slope. According to the findings of previous studies, the slope α was larger for more mature hiPSC-CMs. Therefore, the cause of the variation in α could be attributed to the maturity of the hiPSC-CMs. The pulsation stop temperature was 20.7 ± 2.7 °C, which was consistent with the results of previous studies on mammalian cardiac arrest [27].

According to previous studies, ion channel activity is reduced at low temperatures [26,32,34,35]. Thus, the rate of Ca^2+^ efflux decreases, and Ca^2+^ accumulates in the cell. This causes a delay in excitation–contraction coupling (Ca coupling) and a decrease in the beating frequency. Especially in immature cells, such as hiPSC-CMs, the temperature dependence of ion channels is considered to be directly manifested due to the immaturity of the sarcoplasmic reticulum.

Although pulsation measurements were performed in 11 separate experiments, a larger sample is needed for statistical discussion. For example, in the results of Table 1, the pulsation stop temperature *T*_0_ was estimated to be 20.7 ± 2.7 °C, but this standard deviation might actually have been 1 °C or 4 °C. Nevertheless, there is no doubt that *T*_0_ was more stable than *α* and *β*, where the dependence of the beating frequency f on temperature T was expressed as $f\left(T\right)=\alpha T-\beta$ in the data analysis of this study. Once we have about 100 to 200 samples, we will be able to discuss the distribution of *T*_0_ as well. It will then be possible to discuss in more detail the factors that cause temperature dependence in beating frequency.

All of the hiPSC-CMs used in the experiment were from the same lot. Although all cell groups were seeded on the same day and under the same conditions, the number of cells that adhered to the substrate varied from group to group. This variation may be due to the uneven cell concentration in the culture medium, slight differences in temperature history, or vibration caused by changing the culture medium. One possible way to deal with such cell number variation is to convert the beating power and other factors into a per-cell value. However, it is uncertain whether the beating frequency or its temperature dependence was affected by the number of cells. The time that the hiPSC-CMs were kept under the microscope for pulsatile measurements varied between the cell groups, with the longest ones lasting for more than 3 h.

## 5. Conclusions

We used the cellular measurement system developed in our previous study to measure, in detail, how the beating frequency of hiPSC-CMs changed with temperature. The actual measurement of the beating frequency of hiPSC-CMs revealed that the primitive and universal property of hiPSC-CMs was that they stopped beating at around 20 °C. The mammalian heart also stops beating at around 20 °C, which is due to the cessation of the automatic function of the sinus node [27]. The results of this study indicate that not only did the automatic function of the sinus node, but also the myocardium itself, have a similar temperature dependence. It is possible to measure the iPSC-CMs of animals other than humans using the sensor system used in this study. It is also worthwhile to count the number of cells contributing to the sensor signal in combination with fluorescence microscopy. Furthermore, the feedback control of the piezoelectric stage using sensor signals may enable measurements in a mechanical environment more similar to that of the heart.

## Figures and Tables

**Figure 1 sensors-23-03370-f001:**
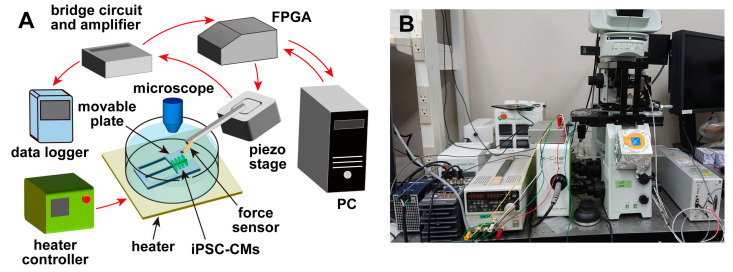
The experimental setup for the pulsation measurements. (**A**) hiPSC-CMs are seeded on a movable plate fixed on the bottom of a petri dish and incubated until they beat; then, they are measured. The displacement sensor can be moved by the piezoelectric stage. The sensor signal is converted to a voltage signal with the bridge circuit, AD-converted in the data logger, and recorded as digital data. (**B**) A photograph of the experimental setup.

**Figure 2 sensors-23-03370-f002:**
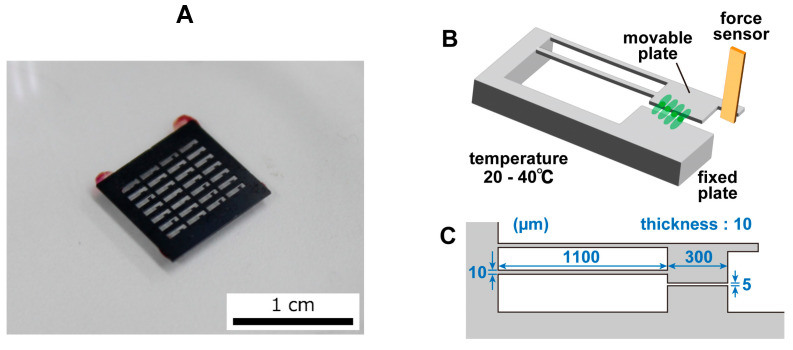
(**A**) The movable plate chip. The 28 movable plates are mounted on one chip. The entire chip is seeded with hiPSC-CMs, and beating cells are selected for measurement. (**B**) Concept of the movable plate. Only the fixed part is attached to the petri dish, and the movable part, which is supported by thin arms, is displaced by the weak force. The force sensor, which is in contact with the protruding part, measures the displacement of the movable part and the force generated by the hiPSC-CMs. In addition, a forced displacement can be applied to the movable plate. (**C**) Dimensions of the movable plate.

**Figure 3 sensors-23-03370-f003:**
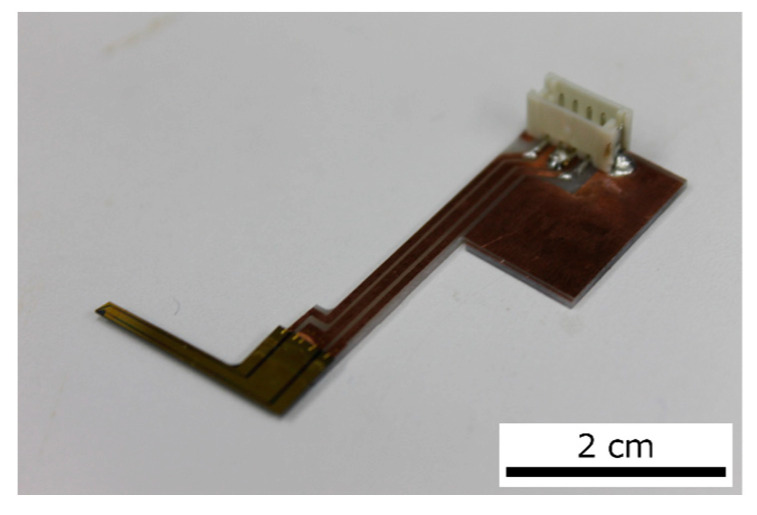
The piezoresistive cantilever and the printed circuit board. The connectors are soldered to the board, and electrical signals are transmitted through these connectors to the bridge circuit.

**Figure 4 sensors-23-03370-f004:**
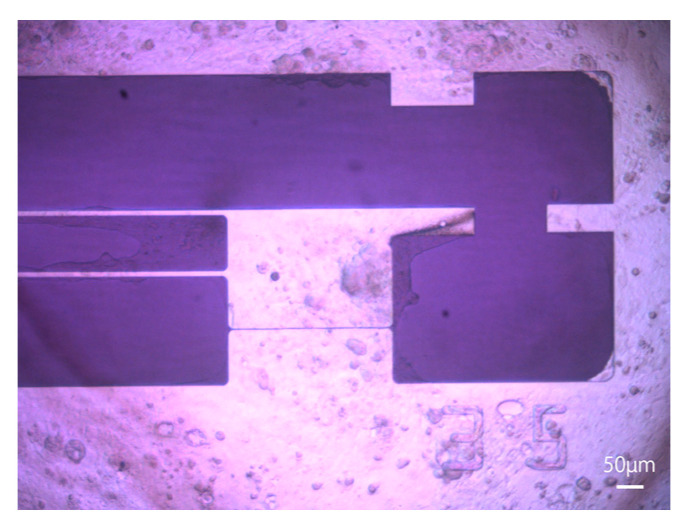
Micrograph of hiPSC-CMs immediately after attachment to a movable plate. After the hiPSC-CMs were cultured for several days, they began to beat. This photograph shows hiPSC-CMs 6 days after seeding.

**Figure 5 sensors-23-03370-f005:**
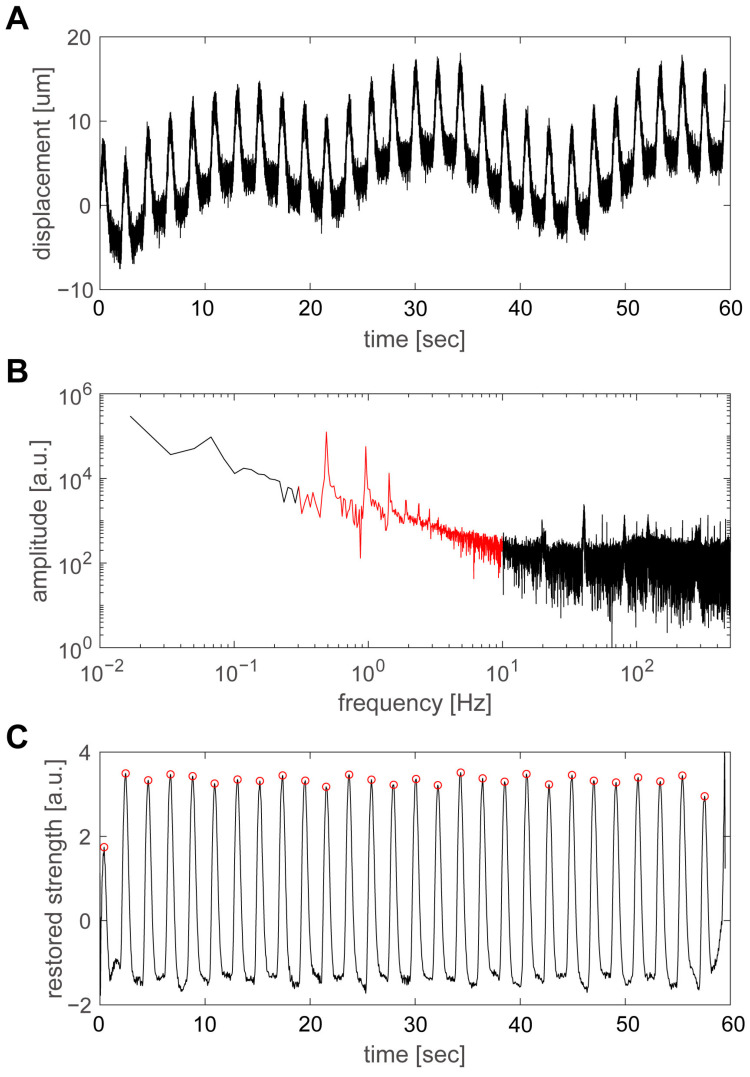
Pulsation of group D at 24 °C. (**A**) The signal (voltage) from the sensor was converted to displacement using the calibration data. (**B**) The spectrum obtained by applying FFT to the sensor signal. The main component of the pulsation, 0.3–10 Hz, is shown in red. (**C**) The time dependence, which was reconstructed by applying inverse FFT to the main component of the pulsation. The peaks of the pulsation were automatically detected and are indicated by the red circles.

**Figure 6 sensors-23-03370-f006:**
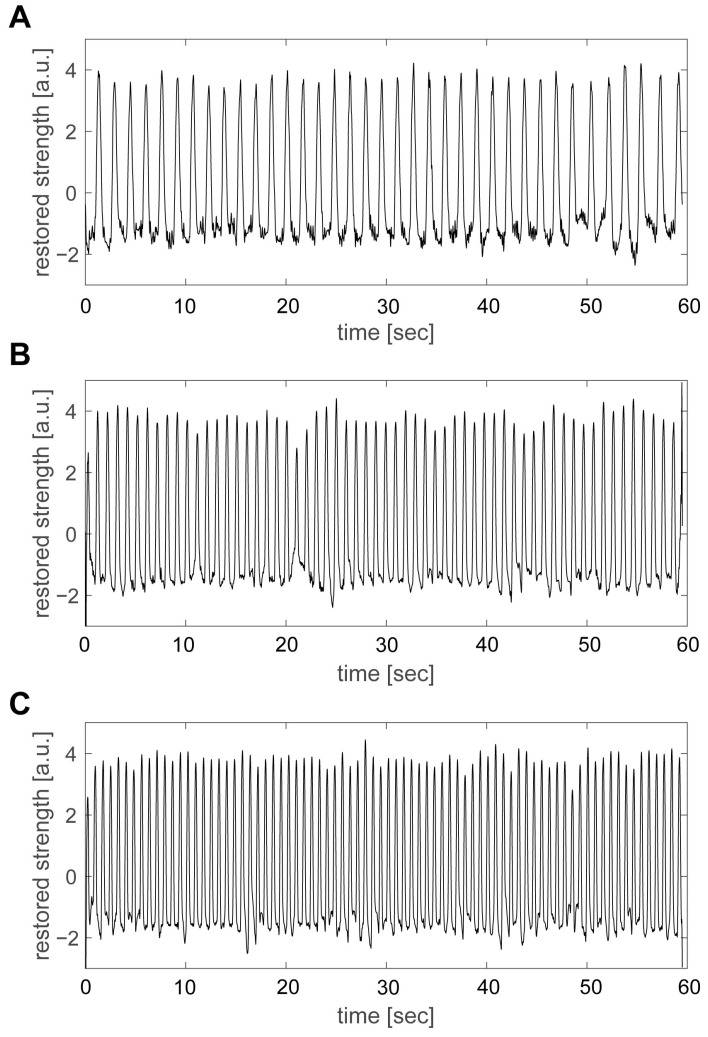
Pulsation of group D at various temperatures. (**A**) 28 °C, 0.5–10 Hz; (**B**) 32 °C, 0.8–10 Hz; (**C**) 36 °C, 0.8–10 Hz. For the pulsation of other cell groups, see the Appendix A.

**Figure 7 sensors-23-03370-f007:**
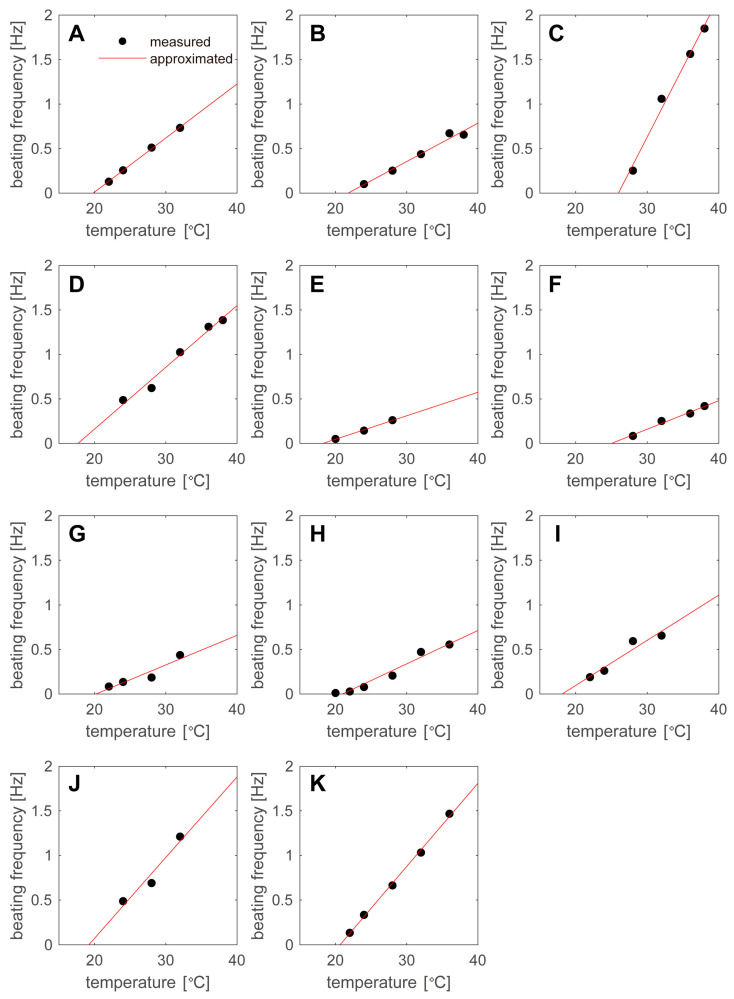
Relationship between temperature and beating frequency. In any hiPSC-CMs, there is a linear relationship between the temperature T (°C) and beating frequency f (Hz). (**A**–**K**) The relationships between temperature and beating frequency for cell groups A to K, respectively.

**Figure 8 sensors-23-03370-f008:**
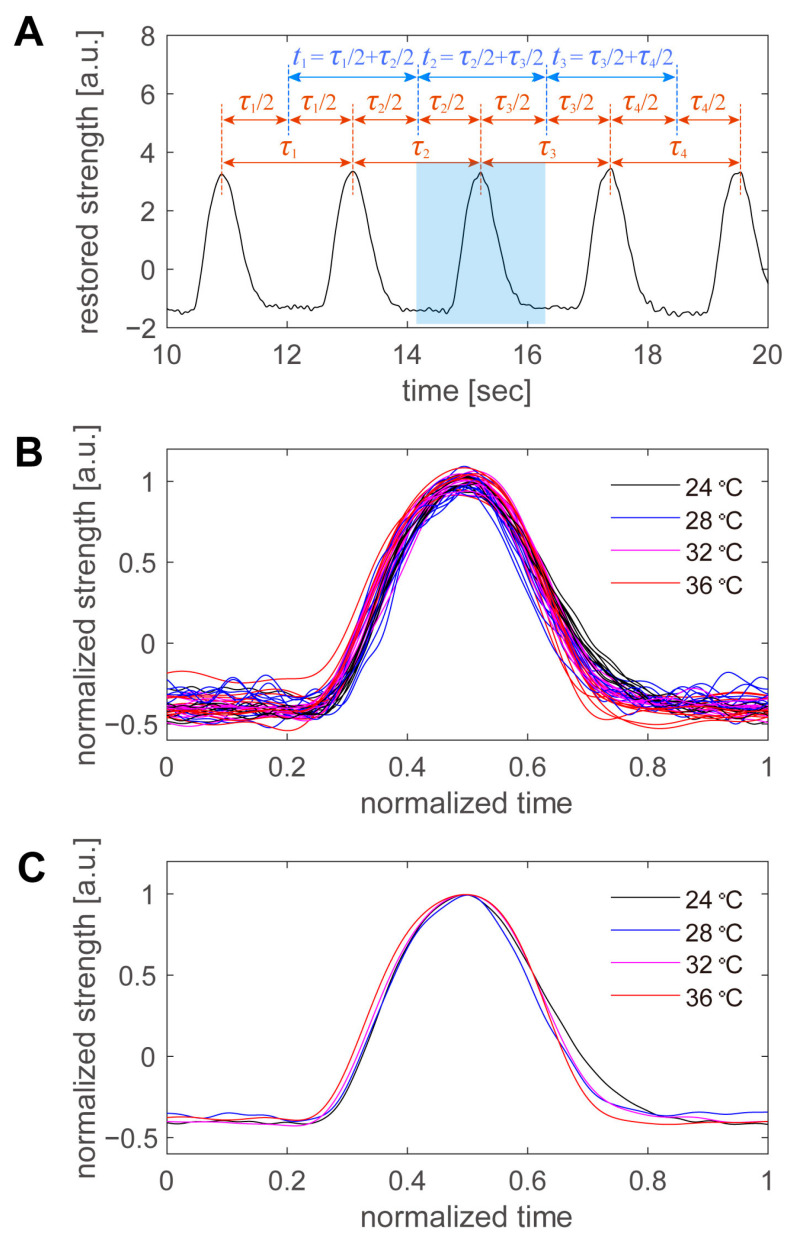
Comparison of the pulsating waveforms at different temperatures. (**A**) Pulsation waveform at 24 °C. A portion of the waveform in Figure 5C was removed. The period of each peak was defined by the boundary of the intermediate time between peaks. (**B**) First 10 beats at each temperature, superimposed over one another. The period of each beat is normalized and displayed. (**C**) Average of the 10 peak waveforms at each temperature.

**Table 1 sensors-23-03370-t001:** Temperature dependence of the beating frequency of hiPSC-CM groups. The coefficients are shown when the temperature is *T* [°C], the beating frequency is *f* [Hz], and the relationship between the two is expressed as $f\left(T\right)=\alpha \left(T-T_{0}\right)$. The pulsation stop temperature *T*_0_ [°C] calculated according to this relationship is also shown.

hiPSC-CM Groups	*α* [Hz/deg]	*T*_0_ [°C]
A	0.0607	19.8
B	0.0431	21.8
C	0.157	26.0
D	0.0694	17.7
E	0.0264	18.2
F	0.0320	25.0
G	0.0333	20.2
H	0.0375	21.0
I	0.0508	18.1
J	0.0904	19.2
K	0.0935	20.6
Average	0.0631	20.7
Standard Deviation	0.0386	2.7

## Data Availability

Not applicable.

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
