# Peer review of "Temperature Dependence of the Beating Frequency of hiPSC-CMs Using a MEMS Force Sensor"

_sensors, 2023, doi:10.3390/s23073370_

Round 1

Reviewer 1 Report

The paper investigates the beating of hiPSC-CMs changes with temperature by using a MEMS force sensor. An experimental setup was demonstrated to measure the pulsation frequency hiPSC-CMs at different environmental temperatures and the corresponding results were obtained.

The authors are requested to respond to the below comments point by point.

 The introduction should be enriched with the most recent research papers (last three to four years) on MEMS based force sensors for similar applications.

·     The dimensions of the force sensor (cantilever) should be included in the manuscript. Whether the force sensor is procured or it is in-house micro fabricated.

·     The specifications of the piezoelectric stage to be included and also the dimensions of the plates shown in figure 3.

·       Please modify the figure 2, showing how the fixed component of the plate is fixed to a petri dish?

·    Since the experiment is carried out in the temperature range of 240C to 400C, whether any temperature compensation is done for the force sensor which uses piezoresistive material for sensing stress in the cantilever with plate displacement.

·      Explain the need of taking a specific 11 cell groups (A to K) and the results shown in figure 6 and 7 are only for group D.

·    The relation between temperature and beating frequency shown in figure 8 are not similar for groups A-D. why?

·      Conclusion can be elaborated with a futuristic application from the results obtained in this study.

Author Response

Please see the attachment. The system seems to allow only one file to be attached. Therefore, we have combined the reply to the Review Report (Reviewer 1), the revised manuscript and the Supplementary Material into one PDF file.

Reviewer 2 Report

The manuscript of “Temperature Dependence of the Beating Frequency of hiPSC-CMs using a MEMS Force Sensor” investigated the temperature dependence of the hiPSC-CMs by integrating the temperature regulation system into a piezo resistive force sensor platform. One of the main challenges to bring hiPSC-CM-related technology into clinical practice is to quantitatively evaluate the degree of maturation of hiPSC-CMs. Accordingly, the authors try to develop a new technique to judge the grade of maturity of hiPSC-CMs base on the theoretical prediction that temperature dependence slope α is larger for more mature hiPSC-CMs according to previous studies. However, the result listed in Table 1 didn’t support this prediction, means failing to evaluate the functional differences between hiPSC-CMs and mature cardiomyocytes by their temperature dependence. Therefore, more efforts should be given to clarify the difference between the theoretical and experiments results.  After the reviewing, the manuscript is recommended to be published in sensors after major revisions. Besides, the following comments should be addressed:

1. The temperature control units should be added in Fig.1a. another concern is that it is difficult to control temperature precisely in the range of 20-40 ℃ in lab only use a heater, without a cooler.

2. Fig. 2 and Fig.3 are suggested to be merged in a single figure.

3. In Fig.5, how many cardiomyocytes were assembled exactly across the gap between the movable and fixed plates? Could the number of cardiomyocytes across the gap effect the collected data like that shown in Fig. 6a?

4. Group C in Table 1 exhibits extra larger α and β than other groups, please discuss.

Author Response

Please see the attachment. The system seems to allow only one file to be attached. Therefore, we have combined the reply to the Review Report (Reviewer 2), the revised manuscript and the Supplementary Material into one PDF file.

Round 2

Reviewer 2 Report

A MEMS-based cantilever force sensor was presented to investigate the temperature dependance of  the beating frequency of hiPSC-CMs. The result reveal that the beating stop at approximately 20.7± 2.7℃. The construction of the sensor as well as the test procedure are interesting.